# Effects of Palmitoylethanolamide on Neurodegenerative Diseases: A Review from Rodents to Humans

**DOI:** 10.3390/biom12050667

**Published:** 2022-05-05

**Authors:** Eugenia Landolfo, Debora Cutuli, Laura Petrosini, Carlo Caltagirone

**Affiliations:** 1Fondazione Santa Lucia IRCCS, 00143 Rome, Italy; eugenia.landolfo@uniroma1.it (E.L.); debora.cutuli@uniroma1.it (D.C.); c.caltagirone@hsantalucia.it (C.C.); 2Department of Psychology, University of Rome “Sapienza”, 00185 Rome, Italy

**Keywords:** PEA, ALIAmides, Alzheimer’s disease, Parkinson’s disease, Huntington’s disease, Frontotemporal dementia, Amyotrophic Lateral Sclerosis, Multiple Sclerosis, neuroinflammation

## Abstract

Palmitoylethanolamide (PEA) stands out among endogenous lipid mediators for its neuroprotective, anti-inflammatory, and analgesic functions. PEA belonging to the N-acetylanolamine class of phospholipids was first isolated from soy lecithin, egg yolk, and peanut flour. It is currently used for the treatment of different types of neuropathic pain, such as fibromyalgia, osteoarthritis, carpal tunnel syndrome, and many other conditions. The properties of PEA, especially of its micronized or ultra-micronized forms maximizing bioavailability and efficacy, have sparked a series of innovative research to evaluate its possible application as therapeutic agent for neurodegenerative diseases. Neurodegenerative diseases are widespread throughout the world, and although they are numerous and different, they share common patterns of conditions that result from progressive damage to the brain areas involved in mobility, muscle coordination and strength, mood, and cognition. The present review is aimed at illustrating in vitro and in vivo research, as well as human studies, using PEA treatment, alone or in combination with other compounds, in the presence of neurodegeneration. Namely, attention has been paid to the effects of PEA in counteracting neuroinflammatory conditions and in slowing down the progression of diseases, such as Alzheimer’s disease, Parkinson’s disease, Huntington’s disease, Frontotemporal dementia, Amyotrophic Lateral Sclerosis, and Multiple Sclerosis. Literature research demonstrated the efficacy of PEA in addressing the damage typical of major neurodegenerative diseases.

## 1. Introduction

### 1.1. PEA, an Anti-Inflammatory and Neuroprotective Substance

Lipid molecules may play a primary role essential to fight, or at least delay, chronic neuroinflammation, a phenomenon underlying many neurodegenerative diseases. A class of anti-inflammatory molecules are the Autacoid Local Injury Antagonist (ALIA) amides [1]. This acronym, coined by the research group of Rita Levi Montalcini, describes a group of endogenous bioactive acyl ethanolamides with anti-inflammatory properties [2], generally referred to as N-acylethanolamines (NAEs). NAEs include PEA, an anti-inflammatory and analgesic substance, oleoylethanolamide (OEA), an anorectic substance, and anandamide (AEA), an endocannabinoid (eCB) substance with autocrine and paracrine signaling properties [3]. PEA cannot strictly be considered a classic eCB, because it has a low affinity for the cannabinoid receptors CB1 and CB2 [4,5]. However, the presence of PEA enhances the AEA activity, likely through an “entourage effect”. PEA is endowed with important anti-inflammatory, neuroprotective, and analgesic actions, and some of its effects are mediated by the peroxisome proliferator-activated receptor (PPAR)-α. PEA anti-inflammatory and neuroprotective functions have been attributed in particular to eCBs belonging to the acyl ethanolamide family, as well as to their congeners, since their production is significantly increased in the sites of neuronal damage [6]. PEA is naturally found in some foods, such as egg yolk, peanut flour, soybean oil, and corn [1,7]. In animal cells, PEA is synthesized from palmitic acid, the most common fatty acid present in many foods including palm oil, meats, cheeses, butter, and other dairy products [8]. Because of its high safety and tolerability [9,10,11,12], PEA is often used as an analgesic, anti-inflammatory, and neuroprotective mediator in the treatment of acute and chronic inflammatory diseases, alone or combined with antioxidant or analgesic molecules acting on molecular targets of central and peripheral nervous system and immune cells [11,13,14]. In the brain, PEA is produced “on demand” by neurons, microglia, and astrocytes, and thus plays a pleiotropic and pro-homeostatic role, when faced with external stressors provoking inflammation. PEA exerts a local anti-injury function by down-modulating mast cell activation and protecting neurons from excitotoxicity [15,16,17]. The synthesis of PEA takes place in the membranes of various cell populations and mainly involves the class of N-acylphosphatidylethanolamines (NAPEs). Similar to its eCB congeners, PEA acts as local neuroprotective mediator and its physiological tone depends on the finely regulated balance between biosynthesis (mainly catalyzed by NAPE-selective phospholipase D) and degradation (mainly catalyzed by fatty acid amide hydrolase (FAAH) and N-acylethanolamine-hydrolyzing acid amidase) [18,19,20].

It was proposed that PEA exerts its effects through three mechanisms, which are not mutually exclusive. The first mechanism advances that PEA acts by down-regulating mast-cell degranulation, via an ALIA effect [21,22,23]; the second one, the entourage effect, postulates that PEA acts by enhancing the anti-inflammatory and anti-nociceptive effects exerted by AEA [4,24,25]; and finally, the third one, the “receptor mechanism”, is based on PEA’s capability to directly stimulate either PPAR-α or the orphan receptor G-protein coupling, GPR55, which mediates many anti-inflammatory effects [26,27].

PEA lacks a direct antioxidant capacity to prevent the formation of free radicals and counteract the damage of DNA, lipids, and proteins [1]. With its lipid structure and the large size of heterogeneous particles in the naïve state, PEA has limitations in terms of solubility and bioavailability. To overcome these problems, PEA has been micronized (m-PEA) or ultra-micronized (um-PEA) [28]. Several in vitro and in vivo preclinical studies attest that PEA, especially in its micrometer-sized crystalline forms, may be a therapeutic agent for the effective treatment of neuroinflammatory pathologies [29]. m-PEA and um-PEA show enhanced rate of dissolution and absorption [30], better bioavailability, pharmacokinetics, and efficacy when compared to its naïve form [31,32]. Since, as already mentioned, PEA has no antioxidant effects per se, the combination of PEA’s ultra-micronized forms with an antioxidant agent, such as a flavonoid, results in more efficacious forms than either molecule alone, potentiating the pharmacological effects of both compounds [33]. In fact, among the natural molecules with excellent antioxidant and antimicrobial functions there are flavonoids, as firstly luteolin, and also polydatin, quercetin, and silymarin. These compounds possess marked antioxidant and neuroprotective pharmacological actions, by modulating apoptosis and release of cytokines and free radicals (reactive oxygen and nitrogen species), suppressing the production of tumor necrosis factor alpha, inhibiting autophagy, and controlling signal transduction pathways [1,34]. In particular, luteolin is able to improve the PEA morphology: while naïve PEA has a morphology featured by large flat crystals, very small quantities of luteolin stabilize the microparticles by inhibiting the PEA crystallization process [35]. The combination of PEA and luteolin makes co-um-PEALut a product able to tackle several neuroinflammatory conditions, and to have protective effects [33].

### 1.2. PEA Action in the Presence of Aging and Neurodegeneration

Aging is the result of a continuous interaction between biological mechanisms and environmental factors, such as life events, health conditions, and lifestyle habits. Although aging is not necessarily synonymous with disease, the deterioration in cell function that increases with advancing age progressively increments the risk of developing disease and disability, because bodily and brain cellular responses become less and less efficient [36]. Namely, aging is characterized by gradual and permanent accumulation of cellular and molecular damage (such as abnormal protein dynamics, mitochondrial dysfunction, DNA damage, oxidative stress, neurotrophin dysfunction), progressive structural changes of neurons (deregulation of neurotransmitters and neuro-signals), loss of tissue and organ function, and neuroinflammatory processes [37,38,39]. Unlike the normally beneficial acute inflammatory response, chronic neuroinflammation can lead to damage and destruction of tissues, and often results from inappropriate immune responses [40]. A fundamental principle behind neuroinflammation is the existence of numerous signaling pathways between glial cells and immune system. Notably, despite different triggering events, a common feature of several central and peripheral neuropathologies is chronic immune activation, particularly of the microglia, the resident macrophages of the central nervous system [41]. Individual neurodegenerative disorders are heterogeneous in etiopathogenesis and symptomatology, but biomedical research has revealed many similarities among them at the subcellular level. These similarities suggest that therapeutic advances against one neurodegenerative disease might ameliorate other diseases as well [42].

The most common neurodegenerative diseases encompass a wide range of conditions which impair mobility, muscle coordination and strength, mood, and cognition. They are amyloidosis, tauopathies, α-synucleinopathies, proteopathies (TAR DNA-binding protein 43, TDP-43), and include Alzheimer’s Disease (AD), Parkinson’s Disease (PD), Huntington’s Disease (HD), Frontotemporal Dementia (FTD), Amyotrophic Lateral Sclerosis (ALS), and Multiple Sclerosis (MS) (Figure 1) [43].

Up to now, the treatment of most of these neurodegenerative diseases was mainly symptomatic (dopaminergic treatment for PD, inhibitors of acetylcholinesterase for cognitive disorders, antipsychotics for dementia), despite significant attempts to find drugs reducing or rescuing the debilitating symptoms [44,45,46]. In this context, integrative treatments of these neurodegenerative diseases have been investigated through a number of in vitro and in vivo animal models of disease, and, when combined with classical drug therapies, are in the frontline of research in an attempt to protect against neuroinflammation and oxidative stress, and thereby improve symptomatology of the neurodegenerative patients [45]. Since most clinical studies on PEA are related to neuropathic pain or inflammation-related peripheral conditions, and there are fewer studies evaluating the possible beneficial effects of PEA on neurodegenerative diseases, we were interested to offer a general overview of the effects of PEA on different symptoms of neurodegeneration, taking into account both human (Table 1) and rodent (Table 2) studies.

## 2. Mild Cognitive Impairment (MCI)

MCI, a neurocognitive disorder often prodromic to dementia, is a cognitive impairment beyond that expected for individual’s age and education, but which does not significantly interfere with activities of daily living [73,74]. There are several subtypes of MCI, which differ based on the type and number of impaired cognitive abilities. When only memory is affected, MCI is defined “amnestic” (a-MCI). When one or more cognitive domains other than memory are affected, MCI is defined “non-amnestic” [75,76].

Only one study evaluated PEA’s effects on MCI symptomatology, describing the efficacy of a 9-month supplementation of co-um-PEALut in an a-MCI patient. While, after 3-months of supplementation, a mild (although non-significant) cognitive amelioration was recorded, at the end of treatment the neuropsychological evaluation reached values almost typical for age and education level, and even the brain SPECT was almost within the normal range [47]. The findings of this case report allow hypothesizing that the use of co-um-PEALut may be a valuable option in the management of MCI-associated neuroinflammation.

## 3. Alzheimer’s Disease (AD)

AD is characterized by progressive and irreversible brain atrophy that spreads unevenly throughout the brain, encompassing particularly vulnerable regions, such as entorhinal cortex, hippocampus, frontal cortex, temporal lobe, and forebrain basal nuclei [77,78,79]. The clinical manifestations are memory loss, cognitive decline, behavioral dysfunction, and inability to carry out the activities of daily living, and seem to be related to the impairment of cholinergic transmission [79,80]. Histopathologically, the main features of AD include extracellular accumulation of amyloid beta (Aβ) fibrils in senile plaques and intraneuronal neurofibrillary tangles (aggregates of the microtubule-associated hyperphosphorylated protein tau) [77].

Several in vitro studies have demonstrated that PEA is able to reduce Aβ-evoked neuroinflammation and attenuate its neurodegenerative consequences. To this purpose, rat primary astrocytes, rat primary mixed neuroglial co-cultures, and organotypic hippocampal slices were challenged with Aβ1-42, and then treated with PEA [81,82,83]. PEA was able to dampen Aβ-induced astrocyte activation and improve neuronal survival through selective PPAR-α activation. Data from organotypic cultures confirmed PEA’s anti-inflammatory properties, implicating PPAR-α mediation, and revealing that the reduction of reactive gliosis induced a marked rebound neuroprotective effect on neurons [82]. Subsequently, the anti-inflammatory and neuroprotective activities of systemically administered PEA were analyzed in adult rats given inflammatory Aβ1-42 intrahippocampal injections mimicking some early traits of AD. By activating PPAR-α, PEA rescued altered molecular pathways and behavioral impairment [57]. Such a capacity of PEA to modulate the protective responses during inflammation suggests that endogenous PEA may be part of the complex homeostatic apparatus controlling the basal threshold of inflammatory processes. Moreover, by blunting Aβ42-induced astrocyte activation, PEA improved neuronal survival [84], cell viability, and glutamatergic transmission in mouse astrocyte-neuron co-cultures [85]. Neuroprotective activities of PEA were studied in mice intracerebroventricularly injected with Aβ25-35, and tested on spatial and non-spatial memory tasks [58]. PEA reduced or even prevented, in a dose-dependent manner, learning, and memory dysfunction. Furthermore, PEA treatment reduced molecular and biochemical inflammatory markers (lipid peroxidation, protein nitrosylation, induction of inducible nitric oxide synthase, and caspase3 activation) [58], counteracted neurodegeneration, and enhanced neuronal viability by dampening reactive astrogliosis and promoting the glial neuro-support function [59]. An additional study provided evidence that subcutaneous treatment with um-PEA is able to exert anti-inflammatory and neuroprotective effects in AD-like mice. Specifically, um-PEA improved learning and memory functions, ameliorated depression and anhedonia, reduced Aβ formation and tau protein phosphorylation, promoted neuronal survival in CA1 hippocampal subregion, normalized astrocytic functions, rebalanced glutamatergic transmission, and restrained neuroinflammation. The um-PEA efficacy was particularly evident in younger mice (6- vs. 12-month-old), suggesting its potential as an early treatment of AD [60]. In the same AD-like model, um-PEA chronic oral administration was capable of rescuing cognitive deficit and decreasing hippocampal level of extracellular glutamate. Once again, the significant neuroinflammatory effect of PEA was found, as shown by the almost complete inhibition of an increase in IL-6 in the hippocampus, by the reduced oxidative stress, and by lower signs of neuronal distress [61,86]. The use of co-ultra-PEALut in an in vitro study also showed interesting results. Pre-treatment significantly reduced inducible nitric oxide synthase and glial fibrillary acidic protein expression, restored neuronal nitric oxide synthase and brain-derived neurotrophic factor, and reduced apoptosis [87]. These results suggest that PEA alone and in combination with lutein may provide an effective strategy for AD.

## 4. Parkinson’s Disease (PD)

PD is a common neurodegenerative disorder primarily characterized by shaking, stiffness, difficulties with balance, walking, and coordination due to the impairment of the dopaminergic nigrostriatal system [88]. Specifically, the loss of dopaminergic neurons projecting from the substantia nigra pars compacta to the caudate-putamen in the striatum results in the loss of dopaminergic neurotransmission, causing the primary motor symptoms [89,90]. Although PD was initially described as a movement disorder without dementia, it is now accepted that the progression of PD affects other dopaminergic, cholinergic, or serotonergic extra-nigral traits, leading to the appearance of cognitive and psychiatric symptoms (i.e., depression and dementia) as well as other non-motor symptoms (i.e., anosmia, sleep disturbances, and constipation) [91]. The ultimate underlying cause(s) of dopaminergic impairment remain(s) unknown [92]. While 5–10% of PD cases are of genetic origin, leading to early PD onset, most cases remain idiopathic and are associated with aging [93], and consequently with neuroinflammation [94]. However, PD must be differentiated from other parkinsonian disorders, including vascular (i.e., striatal infarction), drug-induced (i.e., neuroleptics), metabolic (i.e., Wilson’s disease), infectious (i.e., HIV), toxic (i.e., carbon monoxide), normal pressure hydrocephalus, essential tremor, and other forms of neurodegenerative disease [95].

Given its anti-inflammatory effect, the efficacy of PEA in controlling neurodegeneration associated with neuroinflammation has been evaluated in the animal model of PD induced by 1-methyl-4-phenyl-1,2,3,6-tetrahydropyridine (MPTP) injections. In this model, that exhibits biochemical and cellular changes similar to those observed in idiopathic PD [96], PEA treatment protected against MPTP-induced neurotoxicity. Namely, PEA reduced microglial and astrocyte activation as well as oxidative stress, protected against alterations in protein dynamics in the substantia nigra, and reversed motor deficits. Interestingly, in PPAR-αKO mice treated with MPTP, the lack of PPAR-α exacerbated toxicity of MPTP, indicating that PEA neuroprotection was, at least partially, dependent on PPAR-α [62]. In a more recent study, m-PEA pre-treatment in the MPTP model improved behavioral deficits (measured in open field, pole test, and elevated plus maze), improved tyrosine hydroxylase and dopamine transporter expression (index of PEA protective effect on dopaminergic neurons), reduced pro-inflammatory cytokines, and enhanced hippocampal neurogenesis [63]. These studies reveal that PEA may represent a potential therapeutic candidate to prevent neurodegeneration in the pathogenesis of PD, and propose PEA as a valuable nutraceutical approach to prevent neurodegenerative diseases associated with age. Furthermore, PEA treatment improved the motor disturbances induced by unilateral intra-striatal 6-hydroxydopamine (6-OHDA) injections, increased tyrosine hydroxylase expression at striatal level, reduced expression of pro-inflammatory enzymes, and modulated pro- and anti-apoptotic markers (a valuable index of PEA’s ability to control neuroinflammation and cell death). In addition, PEA provoked a marked protective scavenging and dampened unfolding protein response. Similar data were found in the in vitro studies, where PEA inhibited the damaging stress response of the endoplasmic reticulum [64]. As previously pinpointed, the association of um-PEA and lutein has a strong neuroprotective activity. In fact, co-um-PEALut reduced the specific PD markers (immune-positive tyrosine hydroxylase) and the increased levels in activated astrocytes and pro-inflammatory cytokines, as well as inducible nitric oxide synthase, and modulated the autophagy pathway [65]. More recently, Cordaro and colleagues explored a novel neuroprotective approach based on the use of 2-pentadecyl-2-oxazoline (PEA-OXA). In the MPTP-induced PD model, daily oral PEA-OXA supplementation reduced the behavioral disturbances and neurodegeneration of the dopaminergic tract. Namely, PEA-OXA prevented dopamine depletion, increased tyrosine hydroxylase and dopamine transporter activity, decreased α-synuclein neuronal aggregation, expression of pro-inflammatory enzymes, and inducible expression of nitric oxide synthase. Furthermore, treatment with PEA-OXA significantly limited MPTP-induced microtubule-associated protein-2 alteration, and strongly reduced the increased activation of astrocytes and microglia [66].These pre-clinical studies on experimental models of PD indicate that m-PEA and um-PEA (alone or in association with luteolin or oxazoline) are effective in improving motor functionality through mechanisms aimed at controlling neuroinflammation and neuroprotection, based on an increase in tyrosine hydroxylase and dopamine activity, as well as on the reduction of reactive glia.

As for the human studies, two recent studies highlighted the neuroprotective PEA effect in subjects affected by PD. In particular, in an observational study, Brotini and colleagues demonstrated that oral supplementation with um-PEA slowed disease and disability score progression, significantly reducing most motor and non-motor symptoms [48]. Recently, Brotini has reported the potential efficacy of co-um-PEALut treatment as adjuvant therapy for patients with PD receiving carbidopa/levodopa in treating camptocormia, a postural deformity common in PD in which the spine bends forward while walking or standing. In particular, a 4-month treatment program resulted in complete resolution of the legs and trunk dyskinesia, and marked reduction in camptocormia onset [49]. Therefore, PEA, alone or co-administered and possibly integrating classical treatments, shows therapeutic potential in PD, by correcting dopaminergic deficits and motor dysfunctions, and contrasting pathogenetic aspects involved in the development of the disease.

## 5. Huntington’s Disease (HD)

HD is a rare devastating autosomal dominant disorder, caused by a CAG trinucleotide repeat expansion in the *huntingtin* gene on chromosome 4, which leads to the production of the mutant huntingtin (m-Htt) protein. The degree of symptom severity, disease stage, and markers of neuronal damage correlate with levels of the m-Htt protein in the cerebrospinal fluid in patients with HD [97]. HD is characterized by loss of medium spiny neurons and astrogliosis. The first region to be substantially affected is the striatum, followed by frontal and temporal cortices. The reduced signals from the striatum to the subthalamic nucleus (which sends signals to the globus pallidus) provoke a reduced modulation of movement that results in the characteristic HD movements, called chorea. Besides motor disturbances, HD is characterized by dementia, psychiatric symptoms, and early death. The average age of HD diagnosis is about 40 years, even if the timing of onset is partially determined by the CAG repeats number [98]. Despite its well-defined genetic origin, its molecular and cellular mechanisms are still unclear [99]. There is evidence for a strong correlation between HD and the eCB system, given that the dramatic reduction in cannabinoid receptors in all regions of the basal ganglia is one of the earliest and severe alterations in HD [100].

In R6/2 mice, the transgenic model of HD exhibiting 150 CAG repeats and severe signs of HD, a progressive decline in CB1 receptor expression and abnormal sensitivity to CB1 receptor stimulation are reported. In this model, the relationship was determined between HD symptoms and changes in eCB signaling, measuring eCB levels in brain areas involved in HD (striatum, cortex, and hippocampus) at pre-symptomatic and symptomatic disease stages. Only in symptomatic R6/2 mice were the AEA, 2-Arachidonoylglycerol (2-AG) and PEA levels significantly reduced in the striatum, where cell bodies and dendrites of CB1 receptor-containing neurons are located, indicating that the eCB levels changed according to disease stages and brain regions. Thus, the impairment of the eCB system may represent a hallmark of symptomatic HD. Taking into account its anti-inflammatory and neuroprotective effects, the decreased PEA levels in the striatum of symptomatic R6/2 mice may contribute to the progression of the disease [67]. As mentioned before, with PEA being a substrate for FAAH, it was expected that the use of FAAH blockers could improve HD symptoms in R6/2 mice, suggesting that drugs that inhibit eCB degradation could be used to treat HD.

## 6. Frontotemporal Dementia (FTD)

FTD is a term used for a group of brain disorders characterized by atrophy of the frontal and temporal lobes, and by subsequent behavioral and language disturbances [101]. FTD patients have a characteristic histopathology with cytoplasmic inclusions containing aggregated TDP-43 or tau protein in neurons and glial cells. Signs and symptoms vary depending on which part of the brain is affected. FTD often begins between the ages of 40 and 65 [102].

The only study present in literature on PEA-treated FTD patients reported that a 4-week-treatment with co-um-PEALut reduced behavioral disturbances and improved frontal lobe function in FTD patients through modulation of cortical oscillatory activity and GABAergic transmission [50].

## 7. Amyotrophic Lateral Sclerosis (ALS)

ALS is a fatal neurodegenerative disease characterized by the degeneration of both upper and lower motoneurons and spinal cord which leads to muscle weakness, progressive paralysis, and death within 3–5 years of diagnosis [103,104]. ALS is accompanied by remarkable spinal inflammation mediated in particular by microglia and mast cells. ALS may share some pathological features with FTD, given that ALS patients often show widespread atrophy of the frontotemporal cortices, including premotor cortex [105,106]. ALS is associated with TDP-43 mutations that are observed in more than 90% of ALS patients.

Oral supplementation with um-PEA, in a patient affected by ALS, led to improvement in the clinical picture, muscle tone, respiratory, and motor functions, due to PEA-control of neuroinflammation [51]. More recently, a large clinical trial on ALS patients demonstrated that 6-month-administration of um-PEA, in addition to standard therapy (riluzole), significantly slowed the decline in lung functionality and the worsening of ALS symptoms. This selective PEA effect on diaphragmatic muscle is probably determined by a direct action on the neuromuscular junctions, rather than a nerve-mediated effect leading to regeneration and sprouting of peripheral nerve fibers [52].

## 8. Multiple Sclerosis (MS)

MS is a demyelinating disease with a prominent neurodegenerative component, caused by an autoimmune attack provoking the progressive loss of myelin sheath of neuronal axons [107]. Such a damage decreases the speed of signal transduction, resulting in cognitive and motor impairment depending on lesion’s location. MS progression is caused by episodes of increasing inflammation due to the release of antigens (i.e., myelin oligodendrocyte glycoprotein) [108,109,110]. These antigens elicit a clear autoimmune response that, in turn, sets off a cascade of signaling molecules which result in T cells, B cells, and macrophages crossing the blood–brain barrier and attacking axonal myelin. The attack on myelin starts inflammatory processes that trigger other immune cells and the release of soluble factors, such as pro-inflammatory cytokines. Further release of antigens drives subsequent degeneration, causing further inflammation contributing to the loss of the grey matter. Apart from grey matter and white matter lesions, another distinguishing feature of MS is the presence of undifferentiated oligodendrocyte progenitor cells (OPCs), as a consequence of their inability to progress to a myelin-producing phenotype [111]. Thus, an attractive therapeutic strategy may be to replace lost oligodendrocytes and/or promote their maturation.

Studies indicate that co-um-PEALut may improve the symptoms induced by experimental autoimmune encephalomyelitis (EAE), a MS model. Treatment with co-um-PEALut facilitated the development of undifferentiated and differentiating OPCs, as indicated by the increase in myelin basic protein, proposing this co-compound as a novel approach in the treatment of inflammatory demyelinating disorders, such as MS [40,112,113]. Increasing evidence from animal studies suggests that cannabinoids could efficiently fight demyelination, inflammation, and autoimmune processes occurring in pathologies such as MS [68]. In connection with this, it is reported that in MS the human plasma levels of ECs are altered, the EC system is dynamically modulated depending on the disease subtype [53], and the exogenous administration of eCBs, PEA, and selective inhibitors of eCB re-uptake and hydrolysis improves spasticity, through the enhancement of AEA and possibly 2-AG levels [69]. These studies provide definitive evidence of the tonic control of spasticity exerted by the eCB system, and open new horizons for MS therapy. In the same EAE model of MS, the non-psychoactive effects of cannabidiol (CBD) and PEA, in combination or separately, have been evaluated. Results showed that, whereas CBD-PEA concurrent administration was not as effective as treatment, each drug per se, i.e., single PEA (or CBD) intraperitoneal administration, reduced the severity of neurobehavioral impairment, decreased inflammation, demyelination, cytokine expression, and axonal damage, thus suggesting the effectiveness of PEA as no-psychotropic CB against MS [68]. In the same animal model, intraperitoneal administration of co-um-PEALut significantly reduced the severity of clinical signs. The dose-dependent clinical improvement was associated with the reduced expression of genes coding for inflammatory proteins and the receptors involved in inflammation in the brainstem and cerebellum [70]. In another MS model, the Theiler’s Murine Encephalomyelitis Virus-Induced Demyelinating Disease (TMEV-IDD), PEA administration counteracted motor deficits and exerted an anti-neuroinflammatory effect by reducing the expression of pro-inflammatory cytokines and decreasing microglial activation [71].

As for human studies, a case report described a MS patient with chronic central neuropathic pain who, after adding PEA to their acupuncture treatment, was able to increase the intervals between acupuncture sessions [54]. In a clinical trial on 29 patients with MS, um-PEA oral supplementation relieved pain at the interferon injection site and significantly reduced plasma concentrations of cytokines. Quality of life of the treated patients also improved, when compared with the placebo-treated group [55]. These findings propose that PEA and co-um-PEALut may be the starting point in a novel approach in the treatment of inflammatory demyelinating disorders, as well as in other CNS pathologies classically viewed as primarily neuronal diseases but where myelin and oligodendrocyte loss are relevant.

## 9. Other Diseases (Vascular Dementia, Myasthenia Gravis)

Recent studies revealed that PEA-OXA co-treatment reduced histological changes and neuronal death in vascular dementia induced by bilateral carotid artery occlusion. In particular, this treatment decreased astrocyte markers and microglia activation, increased neuronal development markers, reduced oxidative stress, modulated antioxidant response, and inhibited the apoptotic process [72]. One-week of um-PEA supplementation improved the severity of disability and muscle response to fatigue in patients with myasthenia gravis [56].

## 10. Conclusions

Taken together, the results reported here strongly suggest that by activating multifactorial pharmacological targets and different cellular mediators, PEA could play a promising protective role in counteracting neuroinflammation related to major neurodegenerative diseases. Despite the fact that a simplistic extrapolation of data from the animal model to human condition should be avoided, the results of both preclinical and clinical studies propose PEA as a potential therapeutic agent against neurodegeneration. Mentioned studies emphasize that PEA, especially in its ultra-micronized form, significantly impacts on the progression of some neurodegenerative diseases, acting on specific symptoms and when the pathology is at an early stage. In the final analysis, there is still much to be investigated regarding the effect of PEA, alone or in combination with other compounds, both at the preclinical level and, especially, on subjects with PD, MS, ALS, and FTD. More controversial and less clear is its role in HD. Given the numerous preclinical results regarding the efficacy of PEA in in vitro studies of AD, it would be very interesting to investigate how effective this compound really could be in AD and MCI patients.

In conclusion, through the activation of specific receptors PEA plays a protective role against neuroinflammation related to neurodegeneration, but it is necessary to continue to study the properties of this extraordinary ALIAmide to have a clearer picture on its possible use, also because, to date, no side effects have been reported.

## Figures and Tables

**Figure 1 biomolecules-12-00667-f001:**
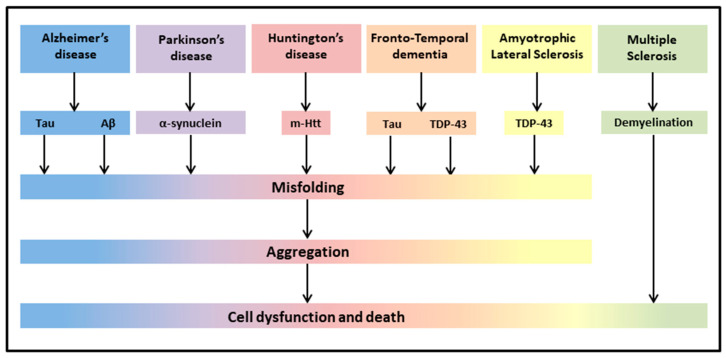
Neurodegenerative diseases share common pathological hallmarks leading to cell dysfunction and death.

**Table 1 biomolecules-12-00667-t001:** Summary of human studies using PEA in the presence of neurodegeneration.

Study	Disease	Sample	um PEA (Alone or In Combination)	Dosage	Duration	Main Outcomes of PEA Treatment
[47]	MCI	1 patient	co-um-PEALut	700/70 mg daily	T3: 3 months treatment T9: 9 months follow-up	T3: mild (though not significant) cognitive improvement; T9: near-normal neuropsychological assessment; improvement in test scores; brain SPECT near-normal.
[48]	PD	30 patients	PEA added to regular levodopa	600 mg daily	12 months	Progressive reduction in the total MDS-UPDRS score; reduction in most nonmotor and motor symptoms.
[49]	PD	1 patient	co-um-PEALut added to regular carbidopa/levodopa	700/70 mg daily	4 months	Complete resolution of leg and trunk dyskinesia and marked reduction in the onset of camptocormia during the “off” state.
[50]	FTD	17 patients	co-um-PEALut	700 mg/2 daily	4 weeks	Improvement in test scores and neurophysiological evaluation; increase in TMS-evoked frontal lobe activity and of high-frequency oscillations in the beta/gamma range.
[51]	ALS	1 patient	PEA	600 mg/2 daily	∼40 days	Improvement in clinical picture.
[52]	ALS	28 treated and 36 untreated patients	PEA + 50 mg riluzole or 50 mg riluzole only	600 mg/2 daily	6 months	Lower decrease in forced vital capacity over time as compared with untreated ALS patients.
[53]	MS	24 patients 17 healthy controls	eCBs levels in blood	_	_	eCB system is altered in MS.
[54]	MS	1 patient	PEA	600 mg/2 daily	∼9 months	Pain reduction; increased interval between acupuncture sessions.
[55]	MS	29 patients	PEA added to IFN-β1a or placebo	600 mg daily	12 months	Improvement in pain sensation, no reduction of erythema at the injection site, improved evaluation of quality of life, increase in PEA, AEA and OEA plasma levels, reduction of interferon-γ, tumor necrosis factor-α, and interleukin-17 serum profile.
[56]	Myasthenia gravis	22 patients	PEA	600 mg/2 daily	1 week	Reduced level of disability and decremental muscle response.

AEA-Anandamide; ALS-Amyotrophic Lateral Sclerosis; co-um-PEALut-combined ultra-micronized PEA/Lutein; eCB-endocannabinoid; FTD-Frontotemporal Dementia; IFN-β1-Interferon-beta-1; MCI-Mild Cognitive Impairment; MDS-UPDRS-Movement Disorder Society-Unified Parkinson’s Disease Rating Scale; MS-Multiple Sclerosis; OEA-Oleoylethanolamide; PD-Parkinson Disease; um-ultra-micronized.

**Table 2 biomolecules-12-00667-t002:** Summary of experimental studies using PEA in the presence of neurodegeneration.

Study	Disease	Sample	um PEA (Alone or In Combination)	Dosage	Duration	Main Outcomes of PEA Treatment
[57]	AD model (Aβ 1–42 intra-hippocampal injection)	Male adult Sprague-Dawley rats (9–12/group)	i.p. PEA PEA added to GW6471	PEA:10 mg/kg; GW647: 2 mg/kg	7 days	Restoration of Aβ 1–42-induced alterations; reduced mnestic deficits.
[58]	AD model (Aβ 25–35 i.c.v. injection)	Male PPAR-α/(B6.129S4-SvJaePparatm 1Gonz) and WT mice (9–10/group)	s.c. PEA and GW7647	PEA: 3–30 mg/kg daily, GW7647: 5 mg/kg daily	1–2 weeks or a single dose	Reduction (10 mg/kg) or prevention (30 mg/kg) of behavioral impairments. No rescue of memory deficits. PEA acute treatment was ineffective.
[59]	AD model	3-month-old male 3 × Tg-AD and WT mice (9–10/group)	s.c. PEA or vehicle	10 mg/kg daily	90 days	Counteraction of disease progression, improvement of trophic support to neurons, in the absence of astrocytes and neuronal toxicity.
[60]	AD model	3-month-old or 9-month-old male 3 × Tg-AD or WT mice (7–11/group)	s.c. PEA or vehicle	10 mg/kg daily	90 days	Improvement of learning and memory, amelioration of depressive and anhedonia-like symptoms, reduced Aβ formation, tau protein phosphorylation, promotion of hippocampal neuronal survival and astrocytic function, rebalancing of glutamatergic transmission, restraint of neuroinflammation.
[61]	AD model	2-month-old male 3 × Tg-AD or WT mice (7–11/group)	oral PEA or vehicle	single dose/sub-chronic/chronic:100 mg/kg daily	1–8–90 days	Rescue of cognitive deficit, restraint of neuroinflammation and oxidative stress, reduced increase in hippocampal glutamate levels.
[62]	PD model (MPTP)	6–7-week-old male PPAR-αKO PPAR-αWT mice (10/group)	i.p. PEA	10 mg/kg	8 days	Reduction of MPTP-induced microglial activation, glial fibrillary acidic protein positive expression astrocyte numbers, overexpression of S100b; protection against alterations in microtubule-associated protein 2a,b, dopamine transporter, nNOS-positive cells in the substantia nigra. Reversal of motor deficits.
[63]	PD model (MPTP)	3/21-month-old male CD1 mice (10/group)	oral PEA	10 mg/kg	60 days	Amelioration of behavioral deficits and of reduction of tyrosine hydroxylase and dopamine transporter in substantia nigra. Reduction of hippocampal proinflammatory cytokines and pro-neurogenic effects.
[64]	PD model (6-OHDA)	Ten-week-old male Swiss CD1 mice (6 × group)	s.c. PEA or GW7647	PEA 3–30 mg/kg/day; GW7647 5 mg/kg/day	28 days	Improvement of behavioral impairment. Increased tyrosine hydroxylase expression at striatal level. Reduction in the expression of pro-inflammatory enzymes, protective scavenging effect.
[65]	PD model (MPTP)	8-week-old male C57BL/6 (10/group)	i.p. co-um-PEALut	1 mg/kg daily	8 days	Reduction of motor impairment, cataleptic response, immobility and anxiety levels. Reduction of neuronal degeneration and of specific PD markers, attenuation of inflammatory processes (activation of astrocytes, pro-inflammatory cytokines, and nitric oxide synthase), stimulation of autophagy.
[66]	PD model (MPTP)	8-week-old male C57BL/6 (10/group)	oral PEA-OXA or vehicle	10 mg/kg daily	8 days	Prevention of MPTP-induced bradykinesia and anxiety, and neuronal degeneration of the dopaminergic tract, prevention of dopamine depletion, modulation of microglia and astrocyte activation.
[67]	HD model	∼32-day-old-R6/2 10-week-old R6/2 mice and WT mice (4/group)	Measurement of PEA, AEA and 2-AG endogenous levels	_	_	Alteration of the eCB system, decreased levels of PEA in the striatum
[68]	MS model (EAE)	12-week-old female C57BL/6 (8/group)	i.p. PEA or CBD or in combination	PEA 5 mg/kg CBD 5 mg/kg	3 days	Reduced severity of EAE neurobehavioral scores, diminished inflammation, demyelination, axonal damage and inflammatory cytokine expression.
[69]	MS model (chronic relapsing EAE)	Biozzi ADH mice (>6/group)	i.v. or i.p. PEA	1–10 mg/kg	Single injection	Amelioration of spasticity
[70]	MS model (EAE)	C57BL/6 mice (8/group)	i.p. co-um-PEALut or vehicle	0.1, 1, and 5 mg/kg	16 days	Dose-dependent improvement of clinical signs through anti-inflammatory signals and pro-resolving circuits.
[71]	MS model (TMEV-IDD)	Four-week female SJL/J mice	i.p. PEA or vehicle	5 mg/kg	10 days	Reduction of motor disability, anti-inflammatory effect.
[72]	Vascular dementia	CD1 mice	Oral PEA-OXA or vehicle	10 mg/kg daily	15 days	Improvement of behavioral deficits, reduction of histological alterations, decrease of markers of astrocyte and microglia activation and oxidative stress, modulation of antioxidant response, inhibition of apoptotic process.

2-AG-2-Arachidonoylglycerol; 6-OHDA-6-hydroxydopamine; Aβ-amyloid beta; CBD-cannabidiol; EAE-Experimental Autoimmune Encephalomyelitis; i.c.v.-intracerebroventricular; i.p.-intraperitoneal; KO-knockout; MPTP-1-methyl-4-phenyl-1,2,3,6-tetrahydropyridine; nNOS-neuronal Nitric Oxide Synthase; PEA-OXA-2-pentadecyl-2-oxazoline; PPAR-α-peroxisome proliferator-activated receptor-α; s.c.-subcutaneous; TMEV-IDD-Theiler’s Murine Encephalomyelitis Virus-Induced Demyelinating Disease; WT-Wild Type.

## Data Availability

Not applicable.

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
