# Peer review of "Effects of Palmitoylethanolamide on Neurodegenerative Diseases: A Review from Rodents to Humans"

_biomolecules, 2022, doi:10.3390/biom12050667_

Round 1
Reviewer 1 Report
The study is devoted to the use of PEA to evaluate its efficacy as a therapeutic agent on neurodegenerative diseases. Neurodegenerative disorders are widespread throughout the world, and although they are numerous and different, they share common patterns of conditions that result from progressive damage to the brain areas involved in mobility, muscle coordination and strength, mood and cognition. The present review is aimed at illustrating in vitro and in vivo researches as well as human studies using PEA treatment, alone or in combination with other compounds, in the presence of neurodegeneration. I liked the review, it is quite informative, but in my opinion additional information is required. I'd like to see a summary table of human trials showing for each trial how many people are included (treatment/control group), what disease, treatment regimen, and outcome. We would like to see the same table in studies involving animals.
Author Response
Dear Editors,
We are submitting the revised version of the paper “Effects of Palmitoylethanolamide on neurodegenerative diseases: A review from rodents to humans” by Landolfo et al.
The paper has been revised according to Reviewers’ requests. In the revised manuscript all changes are underlined.
Point to point comments:
Reviewer 1
“I'd like to see a summary table of human trials showing for each trial how many people are included (treatment/control group), what disease, treatment regimen, and outcome. We would like to see the same table in studies involving animals.”
Following Reviewer 1’s comment, we added 2 Tables summarizing human and experimental trials.
Reviewer 2 Report
Review of a manuscript “Effects of Palmitoylethanolamide on neurodegenerative diseases: A review from rodents to humans” by Eugenia Landolfo and coauthors submitted to “Biomolecules”.
A family of endogenous lipid mediators possess neuroprotective, anti-inflammatory and analgesic functions. Palmitoylethanolamide belongs to the N-acetylanolamine class of phospholipids, which is potentially can be used as a therapeutic agent for neurodegenerative diseases treatment. The authors review recent finding about possible use of palmitoylethanolamide as a remedy for several neurodegenerative disorders. This is an important area of biomedical investigation, and the results of the manuscript will be interesting for the readership of “Biomolecules”.
The following corrections and additions should be made:
Abstract
Line 9: “PEA belongs to the N-acetylanolamine class of phospholipids and was first isolated from soy lecithin, egg yolk, and peanut flour” should be corrected as follows:” PEA belonging to the N-acetylanolamine class of phospholipids was first isolated from soy lecithin, egg yolk, and peanut flour.
Lines 12-13:”The use of PEA, especially in micronized or ultra-micronized forms that maximize its bioavailability and efficacy, has sparked a series of innovative researches to evaluate its efficacy as a therapeutic agent on neurodegenerative diseases.”
This sentence is awkward and should be rewritten as follows: ”The properties of PEA, especially of its micronized or ultra-micronized forms maximizing bioavailability and efficacy, have sparked series of innovative research to evaluate its possible application as a therapeutic agent for neurodegenerative diseases”.
Line 34: ”Palmitoylethanolamide (PEA)” the authors already introduced an abbreviated form on line 8.
Lines 203-205. Parkinson’s Disease (PD)
“PD is a common neurodegenerative disorder primarily characterized by shaking, stiffness, difficulties with balance, walking, and coordination due to the impairment of the dopaminergic nigrostriatal system” The authors should add here a reference on a recent review on PD: ”Biomarkers in Parkinson’s Disease”. Chapter in a book Peplow P.V., Martinez B., Gennarelli T.A. (eds) Neurodegenerative Diseases Biomarkers. 2022. Neuromethods, vol 173. pp 155-180. Humana, New York, NY. https://link.springer.com/protocol/10.1007/978-1-0716-1712-0_7.
Line 386-392: “9. Other neurodegenerative diseases”
In the title the authors write “diseases”, but in the text the give an example of only one disorder - myasthenia gravis. The authors should make an accordance between the title and the text.
Conclusion
Lines 398-402: “Despite the obvious limitations of the mentioned preclinical studies and avoiding any simplistic extrapolation of data from the animal model to the human condition, the results of preclinical experiments propose PEA, especially in its ultra-micronized form, as a potential therapeutic agent, with an impact on the progression of some neurodegenerative diseases, especially on specific symptoms and especially when the pathology is at an early stage.”
This sentence is hard to read. It is first too long and should be split into at least 2 sentences. Besides, using the word “especially” 3 times in one sentence is a bad style.
Author Response
Dear Prof. Cruzzocrea and Dr. Crupi,
We are submitting the revised version of the paper “Effects of Palmitoylethanolamide on neurodegenerative diseases: A review from rodents to humans” by Landolfo et al.
The paper has been revised according to Reviewers’ requests. In the revised manuscript all changes are underlined.
Point to point comments:
Reviewer 2
Abstract: Line 9, Lines 12-13, Line 34: the text has been modified, as required
Lines 203-205: the suggested reference has been inserted
Line 386-392: the title has been modified, as required
Conclusion. Lines 398-402: the sentence has been modified, as required.
Since we added two new Tables, we have deleted the Figure 2. So, in the revised version, the paper has one figure (Figure 1) and two tables (Table 1 and Table 2).
Hoping that the revised version of the paper can meet Reviewers’ criticism, we remain at your disposal for any further modification
Eugenia Landolfo Debora Cutuli Laura Petrosini Carlo Caltagirone
Round 2
Reviewer 1 Report
I have no more remarks/comments on the article. I think that in its present form the article can be recommended for publication.